# Cyberbullying in Gifted Students: Prevalence and Psychological Well-Being in a Spanish Sample

**DOI:** 10.3390/ijerph16122173

**Published:** 2019-06-19

**Authors:** Joaquín González-Cabrera, Javier Tourón, Juan Manuel Machimbarrena, Mónica Gutiérrez-Ortega, Aitor Álvarez-Bardón, Maite Garaigordobil

**Affiliations:** 1Faculty of Education, Universidad Internacional de La Rioja (UNIR), Avenida de la Paz, 137, 26006 Logroño, La Rioja, Spain; joaquin.gonzalez@unir.net (J.G.-C.); javier.touron@unir.net (J.T.); monica.gutierrez@unir.net (M.G.-O.); aitor.alvarez@unir.net (A.A.-B.); 2Faculty of Psychology, University of the Basque Country (UPV/EHU), Avenida de Tolosa, 70, 20018 Donostia, Spain; maite.garaigordobil@ehu.eus

**Keywords:** cyberbullying, gifted, stress, depression, anxiety, health-related quality of life

## Abstract

The differential characteristics of gifted students can make them vulnerable to cyberbullying. There is very little empirical evidence about cyberbullying and giftedness. In the Spanish context, it is unexplored. The main goal of this work is to determine the prevalence of cyberbullying, its distribution in the different roles, and its relationship with other psychological variables. A cross-sectional study was performed with 255 gifted students (M = 11.88 years, SD = 2.28 years) in Spain (155 males, 60.8%). We used the cyberbullying test and the Spanish versions of the DASS-21, ISEL, KIDSCREEN-10, and the SWLS. The results indicate that 25.1% of the students are pure-cybervictims, 3.9% pure-cyberbullies, and 6.6% cyberbully-victims. Pure-cybervictims and cyberbully-victims present worse scores (*p* < 0.001) in health-related quality of life, depression, life satisfaction and stress than the uninvolved individuals. The results suggest that the gifted sample presents more cybervictimization and less cyberbullying than observed in other studies of the general population.

## 1. Introduction

### 1.1. Cyberbullying: Definition, Prevalence, Consequences and Parental Control

Increased access to the internet has been accompanied by an increase of online bullying or cyberbullying. Cyberbullying is defined as conscious and repetitive harm inflicted through the use of computers, mobile phones, or other electronic devices [1]. Together with this definition, there is also the possibility of performing these actions at any time and from any place, as well as the amplifying effect of any action through the social networks [2]. Regarding its prevalence, a Spanish study carried out with a representative sample triangulating the three roles involved in cyberbullying (cybervictim, cyberaggressor, and cyberbystanding) found a 28.2% prevalence of cybervictims, but only 3% of them were pure-cybervictims [3]. In addition, in a survey involving 24 European and Latin American countries, the World Health Organization [4] declared that Spain is one of the countries with the highest rate of cyberbullying. Overall, several studies have concluded that between 20% and 72% of young people have experienced at least one episode of cyberbullying, either as victims or as bullies [2,5,6]. Specifically, a systematic review of Spanish cyberbullying studies [7] found a prevalence of up to 78% for cybervictimization and up to 56% for cyberbullying. 

Cyberbullying is associated with emotional and psychosocial problems and can lead to suicidal ideation [8,9,10]. It is also associated with internalizing and externalizing problems [11], as well as depression [12], stress and anxiety [13,14], and alterations in the cortisol release pattern [15]. Cyberbullying also has an impact on general aspects of subjective well-being such as satisfaction with life or happiness [16,17]. More specifically, there is evidence that the roles associated with victimization and cybervictimization present a significant loss of health-related quality of life (HRQoL) [18,19,20]. Other studies have linked low social support to victimization [21], although few studies have examined this variable in cyberbullying [11]. Moreover, the consequences of cyberbullying increase when these behaviors are maintained over long periods of time [22].

In relation to parental control, studies carried out to date suggest that behavioral control of children constitutes a protective factor against various internet risks [23,24]. Low and Espelage [25] found that parental supervision was associated with less involvement in cyberbullying. Other authors have shown that supervising smartphone usage is associated with a significant reduction in cyberbullying behavior [3]. Similar results have been found regarding various risks related to online grooming of minors (see, for example, Whittle and colleagues [26]). This is in line with the findings that cyberaggression-cybervictimization behaviors are related to family conflicts and poor parent-child communication [27].

### 1.2. Giftedness

Pfeiffer [28], according to his tripartite model, proposes an operational definition of intellectually gifted students in these terms: “(…) compared to others of their age, experience, and opportunities, gifted children are more likely to attain outstanding accomplishments in one or more of the culturally valued domains" (p. 23). Referring specifically to the most academically capable students, he states: "academically gifted students demonstrate outstanding performance, or evidence of potential for outstanding academic performance, in comparison with other students of the same age, experience, and opportunities (...) and a thirst for excelling in one or more areas of academic competition. (...)” [28] (p. 24). Consistent with a developmental approach, we note the growing importance of nonintellectual variables that contribute decisively to establish gifted youngsters’ successful trajectories. In general, gifted students not only show quantitative differences with their age peers, they also show qualitative differences that make them more or less different, such as unusual sensitivity, perfectionism, increased awareness of ethical and moral issues, intense bonds, or early development of internal locus of control, motivation, passion for learning, outstanding academic performance, etc. [29,30]. This sometimes entails becoming a vulnerable and “easier” target for other students who are not different from others. In this sense, some studies have shown that minority groups (for cultural, ethnic, religious or any other reason) are more vulnerable [7,31], drawing on theories like that of social identity [32] and considering that individuals tend to favor their own group (ingroup) while discriminating against others (outgroup). Such distancing from the outgroup has even been found in four- and five-year-olds [33] and could serve as a justification for intimidation simply because a classmate is not a member of the ingroup [34], even encouraging aggression [35]. Therefore, the peculiarities and characteristics of potentially gifted students could increase the odds of them being seen as different, with an outstanding eagerness to learn, exceptional performance, skills that allow them to ask and answer complex questions and excel in schoolwork, etc. This turns them into the target of aggression and bullying, particularly when their strengths do not coincide with the fashionable stereotypes of their age group. This can also occur with above-average students who are not officially identified as gifted, but are recognized by their peers as such [36].

### 1.3. Cyberbullying and Giftedness

Currently, there are very few studies on bullying and giftedness in the international setting (especially in the United States), and the data are contradictory [37]. This was shown by Peterson and Ray [38,39] who found that a high percentage (67%) of gifted students have experienced bullying. On another hand, several studies suggest that this group is similar to the general population in prevalence of bullying [40,41,42] or even presents lower levels of involvement [43,44]. However, there are no current data for Spain or other Spanish-speaking countries.

Studies of gifted student populations are almost non-existent in the field of cyberbullying, as we only found one study of the prevalence of cyberbullying in gifted students. Mitchell [45] found 5% of cybervictims, 4% of cyberbullies, and 17% of cyberbully-victims among gifted students from fifth to eight grade. 

Drawing on this, the main goal of this work is to determine the prevalence of school cyberbullying in a sample of Spanish gifted students and its distribution according to the different roles. Secondary objectives are: (1) to analyze possible differences in relation to the variables sex and age in cyberbullying and the other study variables; (2) to analyze the impact on perceived HRQoL, life satisfaction, depression, and stress of students depending on the cyberbullying role performed; (3) to relate cyberbullying to anxiety, depression, stress, perceived social support, HRQoL, and life satisfaction; and (4) to compare the scores on cybervictimization and cyberbullying depending on general use of the smartphone and parental control over it.

The hypotheses of this study are: (1) the most prevalent role in the sample of gifted students will be that of cybervictim, similar to the general student population [5,7,46]; (2) the percentage of gifted students in the role of cybervictim will be greater than that found in other studies of the general student population [38,39]; (3) differences as a function of sex and age will be similar to those in general student population [3,5,7,47]; (4) students involved in cyberbullying will suffer more psychological impact in the studied variables than students who are uninvolved [8,10,14]; and (5) students who report parental control will obtain lower scores in cybervictimization and cyberbullying than students without such parental control [3].

## 2. Materials and Methods 

### 2.1. Participants

The total number of participants was initially 323 pre-adolescents and adolescents, which, after the initial screening of the database, included 255 adolescents diagnosed as gifted and officially recognized by public institutions (155 males, 60.8%, and 100 females, 39.2%). The mean and standard deviation of age was 11.88 ± 2.28 years with a range of 9–18 years. Concerning educational level, 107 students were in primary education (42.8%), 112 were in compulsory secondary education (44.8%), and 12 were in high school (12.4%).

As in many other countries, the population of gifted students in Spain presents some peculiarities that should be highlighted. Among them, their geographic and school dispersion, the diversity of identification criteria, the lack of systematic detection processes leading to infra-identification, among others [48]. According to the Ministry of Education of Spain, there were 27,133 gifted students in 2017 (last available year). It must therefore be kept in mind that, with reference to the situation of identification in Spain, it is impossible to ensure that the identified students are in any way a representative sample of all the potentially gifted Spanish students. However, the number of students who responded is reasonably large enough to attain a first approximation to the prevalence of giftedness in Spain for this population. If the number of participants who responded to this study had been selected in a strictly random way, the margin of error with a 95% confidence level would be 6.1% for the estimation of proportions.

The sampling was non-probabilistic and incidental. Initially, the researchers contacted 40 associations related to gifted students throughout all the regions of Spain. Of them, 22 agreed to participate, disseminating the study among their affiliates (legal guardians of the surveyed children). The participating associations are scattered around 17 Spanish autonomous communities. We estimate that the documentation was sent to more than 3000 families. In a second phase, we sent the relevant documentation to the legal managers of the associations to send to their members (legal guardians of gifted children) to be able to participate in the study.

### 2.2. Instruments

First, the gifted students responded to questions on sociodemographic variables such as sex, age, grade, association to which their families belonged, and province in which they lived. Other socio-family questions focused on whether their parents limited their use of the smartphone or other devices with internet connection, and whether their parents supervised their actions on these devices. Additionally, we asked about the weekday (Monday to Friday) and weekend usage of the mobile phone.

We used the psychological assessment tools described below:

The Cyberbullying Screening of Peer Harassment (technological scale) [5,49]. 

This tool allows one to triangulate each subject for his/her role in three profiles related to cyberbullying: cybervictim, cyberbully, and cyberbystander. This self-report tool consists of a total of 45 items (15 for each profile) and collects the most significant behaviors associated with cyberbullying. It complies with the recommendations of Berne et al.’s assessment tools review [50], such as providing appropriate indicators of validity (e.g., exploratory or confirmatory factor analyses) and reliability (e.g., dimensional reliability indicators). It uses a 4-point Likert scale (0 = never, 1 = sometimes, 2 = fairly often, and 3 = always), and we coded those who suffered, performed, or observed any cyberbullying behavior "sometimes" as occasional cybervictims, cyberaggressors or cyberbystanders. We considered cyberbullying as severe when the participant suffered, performed, or observed cyberbullying behavior from "fairly often" to "always.” In the study sample, the subscales of cybervictim, cyberbully, and cyberbystander had the following alpha values: α = 0.82, α = 0.72, and α = 0.89, respectively. The Omega coefficient values were, respectively, 0.82, 0.72, and 0.90. Additionally, we added a question addressed to the cyberbystanders, based on the participant role scale (PRS) [51], which allows categorizing the type of cyberbystander as: (a) helps the cyberbully (never starts the bullying, but sometimes participates, supporting the bully; (b) reinforces the cyberbully (sympathizes with the bully, but never directly participates with him/her); (c) uninvolved (is neutral when an aggression occurs); (d) pro-cybervictim (is in favor of the victim, but does nothing to prevent the bullying); and (e) defender (usually actively defends the victim and helps him/her as much as possible). 

The Spanish version of Satisfaction with Life Scale (SWLS) [52]. This 5-item scale evaluates, on a single dimension, the individual’s global judgment of satisfaction with life. It presents adequate indices of reliability and validity. In addition, it is appropriately related to other variables of satisfaction in the school setting, feelings of happiness, and feelings of loneliness. The Cronbach alpha and Omega coefficients were 0.89 and 0.90, respectively, for this sample.

The Spanish version of the Depression, Anxiety, Stress Scales-21 [53]. This self-report assesses negative moods during the past week. It consists of 21 statements that are rated on a 4-point Likert scale, distributed in three subscales with seven items each: depression, anxiety, and stress. In this study, we used the adapted and validated Spanish version by Daza et al. [54]. It has shown evidence of convergent and divergent validity with other self-reports of depression, anxiety, and psychopathology, as well as a three-dimensional solution that empirically supports the proposed subscales [54]. For the subscales of depression, and anxiety, and stress, it presented alpha values of α = 0.90, α = 0.86, and α = 0.91, respectively, in this sample. The Omega coefficient values were, respectively, 0.91, 0.87, and 0.91.

The Spanish version of the Interpersonal Support Evaluation List (ISEL) [55]. This questionnaire assesses respondents’ perceived availability of potential social resources. It consists of four subscales, but we only used the subscales of informative support and sense of belonging. The version used in this research was adapted to Spanish population by Trujillo-Mendoza et al. [56] and it has adequate indices of reliability and validity. The Cronbach alphas for the two dimensions were 0.74 and 0.76, respectively, in this sample. The Omega coefficient values were, respectively, 0.86 and 0.82.

The Spanish version of the KIDSCREEN-10 [57]. This questionnaire assesses HRQoL in children and adolescents from eight to 18 years, from the global perspective of the World Health Organization (WHO), by means of an inclusive (bio-psycho-social) viewpoint. This version presents a global dimension of perceived quality of life. It has suitable indices of reliability and internal validity and is also standardized to Spanish population. The Cronbach alpha and Omega coefficient values were 0.80 and 0.88, respectively, for this sample.

### 2.3. Procedure

This analytical and cross-sectional study was performed between June and October 2017. The measuring instruments were applied to the students online under the supervision of the respective families. The communication process, described in the previous section, was channeled through a specific website created for this study [hidden for reasons of blind review]. We informed the legal guardians about the research team carrying out the project, the purpose of the study, assessment tools, time frame, etc. Before accessing the online form, the legal guardians had to acknowledge that they had been informed of all the above-mentioned points and agree to the participation of their children in the study. The online platform we used was Survey Monkey®. The average time needed to complete the questionnaires was 25 min. The collaboration of the legal guardians and minors was voluntary, anonymous, and disinterested. The project was approved by the Ethics Committee of the Principality of Asturias (Ref.41/17). 

The inclusion criteria for the study were as follows: (a) to have an official diagnosis of giftedness; (b) to belong to one of the associations of giftedness officially registered in Spain, (c) to be of school age and enrolled between fifth grade of primary education and second grade of high school; (d) to have access to the battery of questionnaires prior authorization of the legal guardian. There were no exclusion criteria. As it was an online procedure, and we had no real proof of the exact number of people who had received or read the invitation emails, we cannot cite a specific number of rejections.

### 2.4. Data Analysis

The following analyses were conducted: (1) confirmation of the assumption of normality of the variables involved in the study (Shapiro-Wilks statistic) as well as the variance homogeneity for group comparisons (Levene test); (2) analysis of frequencies and measures of central tendency and dispersion of the instrument; (3) calculation of standardized scores for variables that established relations; (4) chi-square statistic (*χ*^2^) for the contrast of proportions, and Student’s t-test for independent samples for the contrast of mean differences. In cases where statistically significant differences were found, we calculated the Cohen’s d to provide an estimate of the effect size of the difference; (5) partial correlations controlling for age; and (6) analysis of variance with Games-Howel post-hoc comparisons; the value of *p* < 0.05 was considered significant (we calculated the Eta-square (*η*^2^) to provide an estimate of the effect size of the difference). Statistical analyses were carried out using the program Statistical Package for the Social Sciences (SPSS) 23 (IBM®).

## 3. Results

### 3.1. Differences as a Function of Sex and Group

There were no significant differences in the total scores of any cyberbullying dimension as a function of the variable sex: cybervictimization (*t* = 0.893, *p* = 0.373), cyberbullying (*t* = 0.312, *p* = 0.755), and cyberbystanding (*t* = 0.156, *p* = 0.876). Regarding the dimensions of the DASS-21, no differences were obtained according to the variable sex: depression (*t* = 0.258, *p* = 0.797), anxiety (*t* = 0.068, *p* = 0.946), and stress (*t* = 0.521, *p* = 0.603). The same trend applied to the rest of the dimensions: HRQoL (*t* = 0.253, *p* = 0.818), informative support (*t* = 0.419, *p* = 0.832), sense of belonging (*t* = 0.999, *p* = 0.319), and life satisfaction (*t* = 0.510, *p* = 0.978).

There were significant age differences in some dimensions: cybervictimization (*F*_9,243_ = 7.282, *p* < 0.001; η = 0.212), cyberbullying (*F*_9,243_ = 3.837, *p* < 0.001; η^2^ = 0.124), cyberbystanding (*F*_9,243_ = 4.476, *p* < 0.001; η^2^ = 0.142), HRQoL (*F*_9,243_ = 3.729, *p* < 0.001; η^2^ = 0.121), life satisfaction (*F*_9,243_ = 3.070, *p* = 0.002; η^2^ = 0.102), and depression (*F*_9,243_ = 2.270, *p* = 0.019; η^2^ = 0.078). Post-hoc Games-Howel comparisons indicated differences in the prior constructs between the scores of the group of 9-10-year-olds and the 17-18-year-olds (*p* < 0.001), with the latter group obtaining higher mean scores. 

### 3.2. Cyberbullying Profiles and Relationships Between Variables

Next, we calculated the number of pupils involved in the different cyberbullying roles. As can be seen in Table 1, the most prevalent role was that of the cyberbystander (40.8%). On the other hand, 31.8% had suffered cyberbullying behavior, whereas only 10.6% had perpetrated it. We found no significant differences as a function of sex.

In addition, for analytical purposes, we created the following four mutually exclusive categories on the basis of prior categorization: pure-cybervictims (have been occasional or severe cybervictims, but have not perpetrated cyberbullying), pure-cyberbullies (have performed either occasional or severe acts of aggressive behavior, but have never been a victim), cyberbully-victims (have been cybervictims and cyberbullies occasionally or severely), and uninvolved (have not participated either as cybervictim or as cyberbully). The prevalence of these categories was 25.1% (*n* = 64) for pure-cybervictims, 3.9% (*n* = 10) for pure-cyberbullies, 6.6% (*n* = 17) for cyberbully-victims, and 64.3% (*n* = 164) of the sample was uninvolved.

Table 2 shows the various subroles related to the role of the cyberbystander based on the categorization proposed by Salmivalli et al. [51]. There were no significant differences in the distributions of the subroles a function of sex (*χ*^2^ = 4.349, *p* = 0.226).

Table 3 shows the partial correlations (controlling for age) of all dimensions of the study. As can be seen, in general, cybervictimization correlated significantly and negatively with HRQoL (*r* = −0.242, *p* < 0.001) and life satisfaction (*r* = −0.279, *p* < 0.001), and positively with cyberbullying (*r* = 0.432, *p* < 0.001), depression (*r* = 0.325, *p* < 0.001) or stress (*r* = 0.361, *p* < 0.001), among others. 

### 3.3. Differences in Depression, Anxiety, and Stress, Quality of Life, and Social Support as a Function of the Role Performed

Comparisons of the role performed for each of the constructs of study can be seen in Table 4. There were differences in all the constructs (*p* < 0.001) as a function of the four roles. In general, the “uninvolved” role systematically presented the highest scores (high scores in quality of life, sense of belonging, and life satisfaction) and low scores in depression, anxiety, and stress. Cybervictims, and especially cyberbully-victims, had significantly lower scores in quality of life, life satisfaction, and social support, as well as higher scores in depression and stress. It should be noted that, despite there being significant differences in anxiety only between cybervictims and uninvolved, cyberbullies obtained the highest total score.

### 3.4. Use of the Smartphone and Differences in Relation to Parental Control

The results regarding the possession and use of the mobile phone and scores in cybervictimization and cyberaggression are presented in Table 5. It can be seen that children who used the smartphone for five hours per day or more presented significantly higher scores in cybervictimization. These differences were also observed in the case of cyberaggression, although without reaching significance. 

Table 6 shows the results of scores in cybervictimization and cyberaggression as a function of parental control measures (supervision and time limitation). Although there were no significant differences in the socio-family variables, there was a trend in the reduction of cybervictimization and cyberbullying behaviors when parental control was reported. Additionally, we calculated partial correlations (controlling for age) between cybervictimization behaviors and the number of weekday and week-end hours of mobile phone use (*r* = 0.149, *p* = 0.024 and *r* = 0.161, *p* = 0.015, respectively).

## 4. Discussion

The main objective of this study was to analyze the prevalence of cyberbullying in a sample of gifted students. The results showed that 31.5% of the sample is related to the cybervictim profile and 10.6% to the cyberbully profile. This confirms the first hypothesis, that the role of cybervictim is the most prevalent in cyberbullying. These results point in the same direction as the study of Dalosto and Alencar [46], where it was found that gifted students suffered more bullying and performed it less frequently. In addition, when dividing the gifted students into three mutually exclusive categories of involvement, it was observed that almost two thirds of the students who had committed some aggression belonged to the cyberbully-victim category. This suggests that much of the aggression committed may be motivated by prior victimization. Regarding prevalence, the results of our study coincide with other international works like that carried out by Connel et al. [58], who found a prevalence of 25% for cybervictims and of 13.8% for cyberbullies. However, other studies in the general population suggest that the roles of cyberbully-victim [59,60] or of cybervictim-cyberbully-cyberbystander [3] are the most prevalent.

In order to compare the results of this study with those of the general student population, we analyzed various works that used the same tool (The Cyberbullying Test). Garaigordobil [61], in a sample of 3026 participants aged 12 to 18, found that 30.2% of the participants had been cybervictims one or more times in the last year, 15.5% had been cyberbullies, and 65.1% had been cyberbystanders. On the other hand, González-Cabrera et al. [15] found 34.5% of cybervictims, 23.5% of cyberbullies, and 23.5% of cyberbystanders in a sample of students with a mean age of 14.89 years. These data are very similar to those reported in Table 1 in the area of cybervictimization, although they are lower for cyberbullying. However, it is important to note the difference in age between this study (11.88 years mean age) and the afore-mentioned ones (>14 years mean age) because cyberbullying behaviors increase with age, reaching a peak from 14 to 16 years, approximately [62,63]. The most approximate study in age is that of Machimbarrena and Garaigordobil [63], with a mean age of 10.68 years, which reported globally (one or more times in the last year) 13.4% of pure-cybervictims, 0.7% of pure-cyberbullies, and 3.1% of cyberbully-victims. Direct comparison with this study seems to indicate that gifted students present slightly higher prevalence rates in cybervictimization than general student population and also a smaller number of cyberbullies. All this suggests the confirmation of the second hypothesis (the percentage of gifted students in the role of cybervictim will be greater than that found in other studies with general population).

When comparing the results of this study with other studies carried out with different assessment tools in general population, the results of cybervictimization are slightly higher for the gifted group [3,60,64,65]. Studies with representative samples also report lower percentages of cybervictimization, such as the works of Save the Children [66] and González-Cabrera et al. [67]. These data support the second hypothesis of the study (gifted students will present a higher percentage in the role of cybervictim than those found in other studies with general student population).

With regard to the single article on cyberbullying and giftedness that we found, the data contrast with those obtained by Mitchell [45], who suggests that gifted students are less likely to be cybervictims or cyberbullies. The differences between the two works could be due to the fact that Mitchell classified the students based on self-reports (without an external diagnosis by a specialist team) and the time bracket between 2011 and 2018 is very broad for issues related to the internet, as these years have brought about great changes in the possession and use of smartphones, as well as in time spent and use made of the Internet by young people. 

Regarding sex differences, there were no differences in any of the dimensions of cyberbullying. This evidence is shared by several other studies [21,68,69]. However, there were significant differences consistent with findings showing that cyberbullying increases with age [63,69,70]. All this seems to confirm the third hypothesis, which suggested that differences as a function of sex and age would be similar to those of other studies with general student population.

Comparisons among the three involved roles and the uninvolved participants reveal that cybervictims and cyberbully-victims present significantly worse scores than the uninvolved participants in depression, anxiety, stress, sense of belonging and informative support, and HRQoL. All this confirms the fourth hypothesis concerning the greater psychological impact of the roles associated with cybervictimization. These findings are consistent with several studies that found a relation between victimization and depression in the general population [12], cyberbullying and stress and anxiety [13,15], poor quality of life in children suffering from bullying [18,19,20], and low social support [21]. These data are of particular concern for the gifted group of students, because the high prevalence of victimization and its psychological impact occur at an early age. It is important for public collectives and policy makers to take this situation into account, because cybervictimization can generate harmful long-term consequences [71,72].

The results do not yield significant differences in cybervictimization and cyberbullying as a function of parental control, so we cannot confirm the last hypothesis (students who report parental control will obtain lower scores of cybervictimization and cyberbullying than students without such parental control). However, there does seem to be a tendency to present lower mean scores in students who report parents’ limitation of the use of the smartphone and parental control of their actions (see Table 5 and Table 6). Moreover, other studies along these lines suggest that parental control is a protective factor against cyberaggression behaviors [3]. Other authors such as Low and Espelage [25] found that parental supervision was associated with lower involvement in cyberbullying. On the other hand, Ang et al. [23] found that parents’ knowledge of their children’s activities on the internet was associated with less problematic internet use. 

This study has several practical implications: (a) the prevalence found suggests the need to reinforce the strategies of coexistence with students who present an educational singularity like, in this case, giftedness; (b) those responsible for schools that have gifted students enrolled should also ensure that they are adequately integrated and be alert to specific aggressions or bullying, because gifted students are often reluctant to report these experiences [73]; (c) in general, cyberbullying situations may be added to other problems, if any, due to the condition of giftedness, which should be taken into account by the educational teams and the families; (d) reduction of cyberbullying should be included in current existing programs (e.g., Cyberprogram 2.0 [74], "Asegúrate" [Make sure] [75] or Prev@cib [76] in Spain, or modules of specific activities for singular groups such as gifted students. These actions could also be extended to students who excel intellectually, even if they are not officially recognized as gifted [36].

This work also has some limitations. Firstly, the use of self-reports, with the entailed bias of social desirability. This could be improved in the future with complementary measures (reports by parents, teachers, and peers) with which to triangulate a more complete view. A second limitation is the use of cross-sectional methodology and incidental sampling. However, these initiatives in developing sciences as well as the recruitment of difficult-to-access population subgroups through adaptation of sampling strategies are common [74]. In addition, the sampling carried out provides an advantage, as it is a homogeneous convenience sample, which allows for an estimation of the results with clearer generalizability towards the gifted collective [77]. In general, this procedure was the only way to gain access to a largely dispersed population whose localization is complex. Therefore, an equivalent control group would have been extremely complex, if not unfeasible, but this was overcome by employing assessment tools that are widely used in the general population, and that are also standardized. In addition, extrapolation of these results should be done cautiously, as they are to be considered a first approximation to the reality of cyberbullying and giftedness in Spain. Data have been obtained from students who are members of associations, but access to many others who are not members is extremely difficult. Future studies should replicate these findings with additional samples in Spain and in other countries. Students who stand out intellectually over the average but who have no official recognition as gifted should also be incorporated, as a possible line of future research.

## 5. Conclusions

This work provides empirical evidence of a prevalence of 31.6% of cybervictimization in a group of gifted students, even though 38.4% do not own a mobile phone, and the mean age is lower than the habitual age in studies of cyberbullying (focusing on adolescents). These prevalence rates, coupled with an important psychological impact, reveal a situation of vulnerability in this group. Therefore, special attention is required of teachers, who must take this reality into account with regard to support that should be provided to students with specific educational needs. In addition, it is important for institutions and associations linked to these groups to generate plans for cyberbullying prevention and specific actions to improve the situation of this population and the rest of the school population in general.

## Figures and Tables

**Table 1 ijerph-16-02173-t001:** Prevalence of cybervictim, cyberbully, cyberbully-victim and cyberbystander profiles according to the categories of no-problem, occasional, and severe problem (*n* = 255).

		Total *n* = 255 *f* (%)	Males *N* = 155 *f* (%)	Females *n* = 100 *f* (%)	χ^2^	*p*
Cybervictimization	No problem	174 (68.24)	112 (72.26)	62 (62.00)	2.96	0.227
	Occasional	70 (27.45)	37 (23.87)	33 (33.00)		
	Severe	11 (4.31)	6 (3.87)	5 (5.00)		
Cyberbullying	No problem	228 (89.42)	138 (89.03)	90 (90.00)	0.39	0.822
	Occasional	23 (9.01)	15 (9.68)	8 (8.00)		
	Severe	4 (1.57)	2 (1.29)	2 (2.00)		
Cyberbully-victim	No problem	238 (93.33)	145 (93.55)	93 (93.00)	1.57	0.456
	Occasional	16 (6.27)	10 (6.45)	6 (6.00)		
	Severe	1 (0.39)	0 (0.0)	1 (1.00)		
Cyberbystanding	No problem	151 (59.21)	95 (61.29)	56 (56.00)	1.48	0.477
	Occasional	80 (31.47)	48 (30.98)	32 (32.00)		
	Severe	24 (9.42)	12 (7.74)	12 (12.00)		

Note: Cyberbullying: Screening of Peer Harassment [49] can simultaneously assign an adolescent to one or more roles.

**Table 2 ijerph-16-02173-t002:** Distribution of the bystander subroles for the total sample and for each sex.

	Total *n* = 228 *f* (%)	Males *n* = 139 *f* (%)	Females *n* = 89 *f* (%)
Helping the bully	0 (0.00)	0 (0.00)	0 (0.00)
Reinforcing the bully	1 (0.44)	0 (0.00)	1 (1.12)
Uninvolved	36 (15.79)	26 (18.71)	10 (11.24)
Provictim	53 (23.24)	29 (20.86)	24 (26.97)
Defender	138 (60.52)	84 (60.43)	54 (60.67)

**Table 3 ijerph-16-02173-t003:** Means and standard deviations and partial correlations controlling for age between the dimensions of study (*n* = 255).

	1	2	3	4	5	6	7	8	9	10
1. Cybervictimization	-									
2. Cyberbullying	0.43 **	-								
3. Cyberbystanding	0.55 **	0.32 **	-							
4. Depression	0.33 **	0.13 *	0.17 *	-						
5. Anxiety	0.37 **	0.13 *	0.20 **	0.78 **	-					
6. Stress	0.36 **	0.16 *	0.24 **	0.71 **	0.75 **	-				
7. Informative Support	−0.11	−0.12 *	−0.11	−0.21 **	−0.37 **	−0.24 **	-			
8. Sense of Belonging	−0.24 **	−0.17 *	−0.16 *	−0.53 **	−0.50 **	−0.40 **	−0.47	-		
9. Perceived Quality of Life	−0.24 **	−0.18 **	−0.12 **	−0.57 **	−0.57 **	−0.47 **	0.48 **	0.57 **	-	
10. Life Satisfaction	−0.28 **	−0.20 *	−0.10	−0.55 **	−0.68 **	−0.52 **	0.42 **	0.46 **	0.71 **	-
Mean	0.89	0.16	1.59	2.96	2.64	5.10	38.59	20.56	49.95	27.47
*SD*	2.10	0.54	3.13	4.21	3.64	4.87	4.94	3.64	10.95	6.77
Range	0–20	0–5	0–20	0–21	0–21	0–21	22–46	10–28	18.5–83	7–35

Note: * *p* < 0.05. ** *p* < 0.001.

**Table 4 ijerph-16-02173-t004:** Comparisons of the total scores in the depression, anxiety, and stress scale, informative support and a sense of belonging, HRQoL, and life satisfaction based on cyberbullying roles performed by gifted students (*n* = 255).

Roles	Depression		Stress		Anxiety		Inf-Supp		Sense-Bel		HRQoL		Life-Sat	
	*M*	*SD*	*M*	*SD*	*M*	*SD*	*M*	*SD*	*M*	*SD*	*M*	*SD*	*M*	*SD*
UnInvol ^a^	2.12	3.21	3.91	4.02	1.77	2.74	39.30	4.62	21.20	3.44	52.36	10.47	29.22	5.83
Pure-CV ^b^	4.39	5.59	7.08	5.58	4.23	4.66	37.97	5.20	19.59	3.73	46.48	10.97	24.89	7.38
Pure-CB ^c^	3.80	4.24	7.10	6.05	4.40	4.97	37.70	4.14	19.30	3.53	46.58	9.77	25.70	6.43
CB-CV ^d^	5.24	4.66	7.94	5.19	3.94	3.67	34.59	5.48	18.71	3.89	41.76	8.62	21.41	6.74
*F*	6.880 ***		10.321 ***		9.543 ***		5.574 ***		5.339 ***		9.068 ***		13.149 ***	
Post-hoc	b,d > a		b,d > a		b > a		a > d		a > b		b,d < a		b,d < a	

**Note:** UnInvol = uninvolved (*n* = 164); Pure-CV = pure-cybervictims (*n* = 64); Pure-CB = pure-cyberbullies (*n* = 10); CB-CV= cyberbully-victims (*n* = 17); Inf-Supp = informative support; Sense-Bel = sense of belonging; HRQoL = health-related quality of life; Life-sat = life satisfaction; *M* = mean; *SD* = standard deviation; *F* = Welch’s *F*; Post-hoc = Games-Howell; *** *p* < 0.001.

**Table 5 ijerph-16-02173-t005:** Comparison of scores on cybervictimization and cyberaggression as a function of frequency of smartphone use, weekday and weekend hours of use and possession of mobile phone.

	Cybervictimization		Cyberaggression	
	*M (SD)*	*t (p)* *Cohen’s d*	*M (SD)*	*t (p)* *Cohen’s d*
Owns mobile phone with internet				
Yes (*n* = 156)	1.28 (2.47)	4.45 (0.000) *d* = 0.52	0.22 (0.66)	3.03 (0.003) *d* = 0.34
No (*n* = 98)	0.29 (1.02)		0.05 (0.22)	
	**Cybervictimization**		**Cyberaggression**	
	***M (SD)***	***F (p)* η^2^** ***Post-hoc***	***M (SD)***	***F (p)* η^2^** ***Post-hoc***
Frequency of smartphone use				
Daily (*n* = 143) ^a^	1.39 (2.61)	8.05 (0.000) η^2^ = 0.064 c < a	0.20 (0.65)	1.03 (0.360) -
Only on weekends (*n* = 28) ^b^	0.46 (1.04)		0.18 (0.47)	
Rarely (*n* = 69) ^c^	0.23 (0.69)		0.09 (0.33)	
Frequency of mobile use on weekdays				
Less than 1 hour/day (*n* = 131) ^a^	0.43 (1.12)	10.38 (0.000) η^2^ = 0.120 a,b,c < d a < b < d	0.11 (0.36)	2.53 (0.058) -
Between 1 and 2 h/day (*n* = 73) ^b^	1.66 (3.00)		0.19 (0.70)	
Between 3 and 4 h/day (*n* = 24) ^c^	1.21 (2.09)		0.29 (0.75)	
More than 5 ho/day (*n* = 4) ^d^	4.75 (4.57)		0.75 (0.96)	
Frequency of mobile use on weekends				
Less than 1 h/day (*n* = 81) ^a^	0.36 (0.90)	6.63 (0.000) η^2^ = 0.079 a < c,d	0.09 (0.36)	3.08 (0.028) η^2^ = 0.038 a < d
Between 1 and 2 h/day (*n* = 67) ^b^	0.72 (1.45)		0.15 (0.44)	
Between 3 and 4 h/day (*n* = 61) ^c^	1.66 (3.17)		0.18 (0.72)	
More than 5 h/day (*n* = 26) ^d^	1.92 (2.86)		0.46 (0.81)	

Note: *M* = mean; *SD* = standard deviation; *F* = fisher’s *F*; *p* = significance, η^2^ = eta squared; *t* = Student’s *t*.

**Table 6 ijerph-16-02173-t006:** Comparison of scores on cybervictimization and cyberaggression as a function of frequency of parental control (supervision and time limitation).

Limited Time of Smartphone Use	Cybervictimization		Cyberaggression	
	*M (SD)*	*t (p)*	*M (SD)*	*t (p)*
Yes (*n* = 156)	0.77 (1.63)	1.60 (0.114)	0.15 (0.42)	0.67 (0.504)
No (*n* = 82)	1.32 (2.88)		0.21 (0.75)	
Parents supervise smartphone use				
Yes (*n* = 149)	0.87 (2.19)	0.90 (0.367)	0.15 (0.57)	0.71 (0.482)
No (*n* = 85)	1.13 (2.12)		0.20 (0.53)	

Note: *M* = mean; *SD* = standard deviation; *p* = significance, *t* = Student’s *t*.

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
