# Peer review of "Cyberbullying in Gifted Students: Prevalence and Psychological Well-Being in a Spanish Sample"

_ijerph, 2019, doi:10.3390/ijerph16122173_

Round 1

Reviewer 1 Report

This is a very polished paper. Some statistical findings were supportive of previous research. Literature review showed some age as there have been a few international studies that have included intelligence markers, physical appearance, popularity, etc and results have been mixed.

Author Response

REVIEWER COMMENTS FOR THE AUTHOR Reviewer #1:

Reviewer #1: This is a very polished paper. Some statistical findings were supportive of previous research. Literature review showed some age as there have been a few international studies that have included intelligence markers, physical appearance, popularity, etc and results have been mixed.

Authors Thank youfor your general assessment of the manuscript.

Reviewer 2 Report

I enjoyed reading this article and find it helps fill a critical gap in the cyber-bullying literature, and based on my limited knowledge, also addresses some gaps in the literature on vulnerabilities and victimization facing gifted students more broadly.

Having said that, the area in which I think the manuscript could be strengthened is in the critical assessment of the overlap between gifted-ness (conceptually) and issues of identity that also often interact with cyberbullying victimization and offense characteristics. To do this, I would suggest that the authors remark on some of the more nuanced and pressing questions about, for example, race, ethnicity, and access to gifted programs and pre-existing disadvantages for those, say, not in gifted programs but demonstrating some of the same qualities as those who are (which might make them similarly vulnerable).

I think the above suggestion could be integrated in two parts of the paper: the section on gifted-ness and cyberbullying (reviewing the extant literature and understood overlaps) and the discussion. A paragraph in each would probably suffice. To help, here are a couple of sources that might prove useful:

Gifted Education and the Matthew Effect by Leslie Margolin 

Investigating the Intersection of Poverty and Race in Gifted Education Journals: A 15-Year Analysis by Goings and Ford.

Author Response

REVIEWER COMMENTS FOR THE AUTHOR Reviewer #2:

Reviewer #2 I enjoyed reading this article and find it helps fill a critical gap in the cyber-bullying literature, and based on my limited knowledge, also addresses some gaps in the literature on vulnerabilities and victimization facing gifted students more broadly.

Having said that, the area in which I think the manuscript could be strengthened is in the critical assessment of the overlap between gifted-ness (conceptually) and issues of identity that also often interact with cyberbullying victimization and offense characteristics. To do this, I would suggest that the authors remark on some of the more nuanced and pressing questions about, for example, race, ethnicity, and access to gifted programs and pre-existing disadvantages for those, say, not in gifted programs but demonstrating some of the same qualities as those who are (which might make them similarly vulnerable).

I think the above suggestion could be integrated in two parts of the paper: the section on gifted-ness and cyberbullying (reviewing the extant literature and understood overlaps) and the discussion. A paragraph in each would probably suffice. To help, here are a couple of sources that might prove useful:

Authors Thank you very much for your general assessment of the manuscript. We believe that the suggestions are of great interest.

 (1.2 Giftedness) Your contribution about the need to pay attention to those students who, despite presenting characteristics related to high capacities, are not officially identified as gifted and who may be perceived by their peers as such (therefore, becoming potential victims) has been included in the introductory section.

Due to its interest, this is also discussed and added for prospective future work.

1.2 Giftedness (lines  95-96)

... This can also occur with above-average students who are not officially identified as gifted, but are recognized by their peers as such [36].

Discussion (lines 399-400)

…These actions could also be extended to students who excel intellectually, even if they are not officially recognized as gifted [36].

On the role of the ethnic group in the context of the diagnosis of giftedness, we believe that it is not necessary to mention it in the introduction. The reason is that the whole sample is ethnically similar. In countries like the USA, it is common to find these situations. However, although Spain is increasingly ethnically pluralistic, all participants are Caucasian/white and of Spanish nationality.

Reviewer 3 Report

The topic of the paper is interesting and highly relevant. But I think the authors should make an effort to simplify it. There are too many measurements that are insufficiently discussed. Why use both life-satisfaction and perceived quality of life? If there is a clear reason then that needs to be discussed.

There is nothing in the introduction about parental control. It just shows up in the hypothesis at the very end. Since this is one of the key concept it needs to be introduced properly earlier in the introduction.

It varies how many numbers there are after a decimal point throughout the manuscript. Sometimes there is one, other time two and sometimes none. This should be harmonized. One number is usually preferable.

Table 2 is unfinished. There are percentages by some numbers and not by others. And again, the number of decimals is inconsistent.

I’m not sure that table 3 is the best way of presenting the authors case. It groups the cyberbullying variables with the more “dependent” variables.

Table 4 is overcrowded with numbers in the current setup.

I have a hard time understanding table 5. This needs to be reformatted or better explained.

Minor comment:

Line 24: The sentence is long. I recommend splitting it “…cyberbullying and giftedness. In the Spanish context, it is unexplored.“

Line 28: Recommend: “(M=11.9 years, SD=2.3 years) in Spain (155 males, 60.8%).

Line 34: Recommend „…other studies of the general population“.

Line 59: Recommend “…domains" (p. 23). Referring specifically to the most academically-capable…“

Lines 63-64: Recommend: „Consistent with a developmental approach, we note the growing importance of nonintellectual…“

Line 69: I have great problems with the use of „legimate“ in this context.

Line 74: I have great problems with the use of „justify“ in this context.

Line 77: “…different from those of their environment…“ Unclear phrasing.

Lines 77-78: It‘s not good to repeat the word „outstanding“ twice in the same sentence. Find a synonym.

Line 79: Skip the “s” in “aggression”

Lines 84-85: Recommend “This was shown by Peterson and Ray…“

Lines 89-90: This sentence needs to be rephrased. There are plenty of studies on cyberbullying but it is true that there are not many on cyberbullying among gifted students.

Line 103: Recommend “…in other studies of the general…”

Line 116: Recommend “…and 12 were in high school”

Lines 120-121: Recommend “…were 27,133 gifted students in 2017…“

Line 121: Recommend: “…kept in mind that, with reference to the situation…”

Line 144: Eliminate the space before the full stop.

Line 148: Add space after comma.

Line 167: Coefficients

Lines 168, 177 and 184: “The Spanish version…”

Line 184: Comma after WHO

Line 190: Recommend “…June and October 2017.“

Line 194: Promoters is not right. Do you mean Organizers? 

Line 195: Recommend: “…study, assessment tools…“

Line 197-198: Recommend “The online platform we used was Survey Monkey®. The average time needed to complete the questionnaires was 25 minutes.“

Line 207: Correct Data anaylsis

Line 223: no differences

Lines 247-271 bleed into the text. The editors need to correct this.

Line 268: subroles

Line320-321: Recommend “This confirms the first hypothesis, that the role of cybervictim is most prevalent in cyberbullying.“

Line 326: Recommend “…by prior victimization“

Line 333: Recommend “various work that used the same tool“

Line 363: Omit “Regarding sex differences…“ Otherwise the sentence becomes repetitive.

Line 376-378: Chronification is not a word. Recommend shortening to “It is important for public collectives and policy makers to take this situation into account because cybervictimization can generate harmful long-term consequences.

Line 386: On the other hand

Line 390: I have no idea what the authors are trying to say here: “…reinforce the strategies of coexistence in the political-educational field with students“

Line 393: “Denounce“ is not the right word

Line 418: Recommend shortening the beginning to “This work provides….“

Line 422-424: Recommend: “Therefore, special attention is required of teachers, who must take this reality into account with regard to support that should be provided to students with specific educational needs.“

Line 425: Again, you need a better term – maybe just delete ‘political-educational’

Author Response

REVIEWER COMMENTS FOR THE AUTHOR Reviewer #3:

Reviewer #3 The topic of the paper is interesting and highly relevant. But I think the authors should make an effort to simplify it. There are too many measurements that are insufficiently discussed. Why use both life-satisfaction and perceived quality of life? If there is a clear reason then that needs to be discussed.

Authors Thank you for your general assessment of the manuscript and the relevance of the study topic.

The authors wanted to use a wide variety of instruments in which to be able to evaluate the psychological influence as related to cybervictimization.

It is true that some of these constructs are not sufficiently well developed in the introduction, and some changes have been incorporated.

However, it is important to note that the use of life-satisfaction and HRQoL is complementary. Life-satisfaction is an indicator of the general subjective well-being that is not contemplated in the HRQoL (as it is evaluated with the KIDSCREEN-27).

Cyberbullying is associated with emotional and psychosocial problems and can lead to suicidal ideation [8–10]. It is also associated with internalizing and externalizing problems [11], as well as depression [12], stress and anxiety [13,14], and alterations in the cortisol release pattern [15]. Cyberbullying also has an impact on general aspects of subjective well-being such as satisfaction with life or happiness [16,17]. More specifically, there is evidence that the roles associated with victimization and cybervictimization present a significant loss of health-related quality of life (HRQoL) [18–20]. Other studies have linked low social support to victimization [21], although few studies have examined this variable in cyberbullying [11]. Moreover, the consequences of cyberbullying increase when these behaviors are maintained over long periods of time [22].

Reviewer #3 There is nothing in the introduction about parental control. It just shows up in the hypothesis at the very end. Since this is one of the key concept it needs to be introduced properly earlier in the introduction.

Authors Thank you very much for your suggestion. This issue was addressed in the introduction, but we acknowledge that greater depth was needed. We have redrafted the manuscript, adding some more citations.

In relation to parental control, studies carried out to date suggest that behavioral control of children constitutes a protective factor against various Internet risks [23,24]. Low and Espelage [25] found that parental supervision was associated with less involvement in cyberbullying. Other authors have shown that supervising smartphone usage is associated with a significant reduction in cyberbullying behavior [3]. Similar results have been found regarding various risks related to online grooming of minors (see, for example, Whittle and colleagues [26]). This is in line with the findings that cyberaggression-cybervictimization behaviors are related to family conflicts and poor parent-child communication [27].

Reviewer #3 It varies how many numbers there are after a decimal point throughout the manuscript. Sometimes there is one, other time two and sometimes none. This should be harmonized. One number is usually preferable.

Authors We decided to unify the entire manuscript and always present two decimals.  This is important, especially in values that are between 0 and 1, as in correlation coefficients. In the case of significance, the three decimal places have been maintained, as is usual in our field of research.

Reviewer #3 Table 2 is unfinished. There are percentages by some numbers and not by others. And again, the number of decimals is inconsistent.

Authors We have corrected this table, both in the number of decimals and in its format.

Reviewer #3 I’m not sure that table 3 is the best way of presenting the authors case. It groups the cyberbullying variables with the more “dependent” variables.

Authors The table has maintained the same format as it may be of interest to compare the scores between different variables

Reviewer #3 Table 4 is overcrowded with numbers in the current setup.

Authors -> Several changes have been made to the table to make it easier to read and less crowded.

Reviewer #3 I have a hard time understanding table 5. This needs to be reformatted or better explained.

Authors -> Table 5 has been divided into two tables for better comprehension. In addition, a slightly more extensive interpretation has been made of the two tables.

Minor comment:

Author -> Thank you very much for the detailed level of feedback. All the issues raised have been taken into account and corrected.

Reviewer #3 Line 24: The sentence is long. I recommend splitting it “…cyberbullying and giftedness. In the Spanish context, it is unexplored.“

Line 28: Recommend: “(M=11.9 years, SD=2.3 years) in Spain (155 males, 60.8%).

Line 34: Recommend „…other studies of the general population“.

Line 59: Recommend “…domains" (p. 23). Referring specifically to the most academically-capable…“

Lines 63-64: Recommend: „Consistent with a developmental approach, we note the growing importance of nonintellectual…“

Line 69: I have great problems with the use of „legimate“ in this context.

Line 74: I have great problems with the use of „justify“ in this context.

Line 77: “…different from those of their environment…“ Unclear phrasing.

Lines 77-78: It‘s not good to repeat the word „outstanding“ twice in the same sentence. Find a synonym.

Line 79: Skip the “s” in “aggression”

Lines 84-85: Recommend “This was shown by Peterson and Ray…“

Lines 89-90: This sentence needs to be rephrased. There are plenty of studies on cyberbullying but it is true that there are not many on cyberbullying among gifted students.

Line 103: Recommend “…in other studies of the general…”

Line 116: Recommend “…and 12 were in high school”

Lines 120-121: Recommend “…were 27,133 gifted students in 2017…“

Line 121: Recommend: “…kept in mind that, with reference to the situation…”

Line 144: Eliminate the space before the full stop.

Line 148: Add space after comma.

Line 167: Coefficients

Lines 168, 177 and 184: “The Spanish version…”

Line 184: Comma after WHO

Line 190: Recommend “…June and October 2017.“

Line 194: Promoters is not right. Do you mean Organizers?

Line 195: Recommend: “…study, assessment tools…“

Line 197-198: Recommend “The online platform we used was Survey Monkey®. The average time needed to complete the questionnaires was 25 minutes.“

Line 207: Correct Data anaylsis

Line 223: no differences

Lines 247-271 bleed into the text. The editors need to correct this.

Line 268: subroles

Line320-321: Recommend “This confirms the first hypothesis, that the role of cybervictim is most prevalent in cyberbullying.“

Line 326: Recommend “…by prior victimization“

Line 333: Recommend “various work that used the same tool“

Line 363: Omit “Regarding sex differences…“ Otherwise the sentence becomes repetitive.

Line 376-378: Chronification is not a word. Recommend shortening to “It is important for public collectives and policy makers to take this situation into account because cybervictimization can generate harmful long-term consequences.

Line 386: On the other hand

Line 390: I have no idea what the authors are trying to say here: “…reinforce the strategies of coexistence in the political-educational field with students“

Line 393: “Denounce“ is not the right word

Line 418: Recommend shortening the beginning to “This work provides….“

Line 422-424: Recommend: “Therefore, special attention is required of teachers, who must take this reality into account with regard to support that should be provided to students with specific educational needs.“

Line 425: Again, you need a better term – maybe just delete ‘political-educational’